# Optimizing Quality and Shelf-Life Extension of Bor-Thekera (*Garcinia pedunculata*) Juice: A Thermosonication Approach with Artificial Neural Network Modeling

**DOI:** 10.3390/foods13030497

**Published:** 2024-02-04

**Authors:** Shikhapriyom Gogoi, Puja Das, Prakash Kumar Nayak, Kandi Sridhar, Minaxi Sharma, Thachappully Prabhat Sari, Radha krishnan Kesavan, Maharshi Bhaswant

**Affiliations:** 1Department of Food Engineering and Technology, Central Institute of Technology, Kokrajhar 783370, India; gshikhapriyom@gmail.com (S.G.); pdas12994@gmail.com (P.D.); pk.nayak@cit.ac.in (P.K.N.); 2Department of Food Technology, Karpagam Academy of Higher Education (Deemed to Be University), Coimbatore 641021, India; 3Department of Applied Biology, University of Science and Technology Meghalaya, Baridua 793101, India; minaxi86sharma@gmail.com; 4Department of Food Science and Technology, National Institute of Food Technology, Entrepreneurship and Management, Kundli 131028, India; sari.tp.tp@gmail.com; 5New Industry Creation Hatchery Center, Tohoku University, Sendai 980-8579, Japan; 6Center for Molecular and Nanomedical Sciences, Sathyabama Institute of Science and Technology, Chennai 600119, India

**Keywords:** bor-thekera, juice, thermosonication, microbial activity, preservation, ANN, antioxidant activity

## Abstract

This study investigated the quality characteristics of pasteurized and thermosonicated bor-thekera (Garcinia pedunculata) juices (TSBTJs) during storage at 4 °C for 30 days. Various parameters, including pH, titratable acidity (TA), total soluble content (TSSs), antioxidant activity (AA), total phenolic content (TPC), total flavonoid content (TFC), ascorbic acid content (AAC), cloudiness (CI) and browning indexes (BI), and microbial activity, were analyzed at regular intervals and compared with the quality parameters of fresh bor-thekera juice (FBTJ). A multi-layer artificial neural network (ANN) was employed to model and optimize the ultrasound-assisted extraction of bor-thekera juice. The impacts of storage time, treatment time, and treatment temperature on the quality attributes were also explored. The TSBTJ demonstrated the maximum retention of nutritional attributes compared with the pasteurized bor-thekera juice (PBTJ). Additionally, the TSBTJ exhibited satisfactory results for microbiological activity, while the PBTJ showed the highest level of microbial inactivation. The designed ANN exhibited low mean squared error values and high R^2^ values for the training, testing, validation, and overall datasets, indicating a strong relationship between the actual and predicted results. The optimal extraction parameters generated by the ANN included a treatment time of 30 min, a frequency of 44 kHz, and a temperature of 40 °C. In conclusion, thermosonicated juices, particularly the TSBTJ, demonstrated enhanced nutritional characteristics, positioning them as valuable reservoirs of bioactive components suitable for incorporation in the food and pharmaceutical industries. The study underscores the efficacy of ANN as a predictive tool for assessing bor-thekera juice extraction efficiency. Moreover, the use of thermosonication emerged as a promising alternative to traditional thermal pasteurization methods for bor-thekera juice preservation, mitigating quality deterioration while augmenting the functional attributes of the juice.

## 1. Introduction

Bor-thekera (*Garcinia pedunculata*) is a fruit widely found in the northeastern part of India, including Assam, Arunachal Pradesh, Meghalaya, and Manipur. In Asia, its distribution is mostly seen in Bangladesh and India. The fruit is fleshy, smooth, and globose with 7–9 subequal sections of flesh. The fruit is 6–11 cm in diameter and contains 8–10 brown seeds. Each of the seeds is surrounded by slimy, juicy orange-colored pulp. The ripe fruit has a highly sour taste. Ripe fruit is orange-yellow and used by ethnic people for its numerous benefits. It is well known for treating asthma, cough, bronchitis, dysentery, digestion, and fever. It was found to contain hydroxycitric acid, garcinol, and cambogin. It has antioxidant, antimicrobial, antidiabetic, hepatoprotective, and neuroprotective activities [1]. It can improve iron and high-density lipoprotein cholesterol levels while lowering serum alkaline phosphatase, alanine transaminase, lactate dehydrogenase, and glucose levels in the human body [2].

Bor-thekera is a seasonal fruit, typically available from February to April. It is commonly consumed either in its raw state or in processed forms, such as jams or juices. The fruit is very perishable in nature; therefore, it needs to be preserved to increase its availability throughout the year. Mostly, it is preserved by drying (slices of bor-thekera fruit) and making pickles, jam, and jelly for later use. But in these forms, most of the bioactive components and organoleptic aspects of the fruit are destroyed. Therefore, an effective way of preserving bor-thekera fruit could be in the form of juice to retain the maximum level of bioactive compounds. Moreover, the consumer demand for fresh-like quality could be achieved.

Pasteurization is the most popular heat treatment for eliminating pathogenic microbes and deactivating enzymes in fruit juices but it may result in unfavorable alterations to the physiochemical, biological, and organoleptic aspects, as the juices are treated at temperatures above 80 °C [3]. The effectiveness of microbial destruction and the effect on quality attributes are influenced by the food matrix [4] present during pasteurization. Employing novel processing methods can be considered an alternative, eco-friendly approach to enhance the quality attributes of juices for sustainable food practices [5].

In recent times, ultrasound has now been demonstrated to be efficient against pathogenic microorganisms in liquid foods and can meet the US Food and Drug Administration’s 5-log reduction criterion for some contaminating pathogens, like *Escherichia coli* [6]. Using ultrasound alone presents several limitations, with its effectiveness hindered by prolonged treatment times and high energy requirements, making it unsuitable as a potential substitute for thermal treatment. Therefore, it is imperative to acknowledge that ultrasound, on its own, may not consistently achieve the desired inactivation targets and may pose challenges in terms of feasibility for upscaling. To address these limitations, a strategic approach involves combining ultrasound with heat (50 °C), namely, thermosonication (TS), which offers a promising solution that can overcome the identified problems and enhance the overall efficacy of the treatment process (50 °C) [7]. TS is an integrated method of processing with mild heat treatment (≤60 °C) along with low-frequency sonication (20–100 kHz), which provides a processing effect similar to pasteurization in a shorter time (15–60 min) with very minute changes in nutritional qualities [8]. When heat and ultrasound are applied together in thermosonication, it leads to the destruction of microbial cell walls, as the sensitivity of microbial cells increases immensely [9]. Ultrasound causes a significant quantity of cavitations because of the extra heat, and therefore, has more of an impact on microorganism destruction, as well as enzyme inactivation. Hence, it reduces the processing time by fifty-five percent and temperature by sixteen percent [10]. When heat is incorporated into the treatment, it has a dual effect on the cavitation bubbles. On one hand, the bubbles may become smaller due to the increased thermal energy. On the other hand, the elevated temperature renders microbial cells more susceptible to the mechanical stresses caused by microstreaming and shock waves. This synergistic interaction between ultrasound-induced cavitation and elevated temperature results in a more effective microbial inactivation process. The combination of smaller bubbles and increased temperature enhances the overall efficiency of the treatment, leading to higher microbial inactivation rates.

The use of thermosonication for the processing of juices from fruits such as tangerine, pomegranate, elephant apple, blueberry, pineapple, orange, amora, and quince showed better retention of quality parameters [11,12,13,14,15,16,17].

Artificial neural network optimization is an effective approach that allows for the use of neural networks’ capabilities to find the best responses to intricate issues. The human brain serves as a basis for neural networks, which are composed of interconnected nodes (neurons) that process and send information [5]. Neural networks are frequently used as function approximators to optimize various settings. The objective of the study was to train the ANN on the fundamental relationship between an input variable and the corresponding expected output [3]. The network can be trained to make accurate predictions or generate the outputs that are required by changing its weights and biases. Therefore, this study focused on the quality attributes and microbial activity of pasteurized and thermosonicated bor-thekera (*Garcinia pedunculata*) juices (TSBTJs), which were assessed during one month of storage at 4 °C. Moreover, a multi-layered artificial neural network (ANN) was employed to both model and optimize the ultrasound-assisted extraction process for bor-thekera juice, focusing on physico-chemical attributes.

## 2. Materials and Methods

### 2.1. Juice Preparation

Bor-thekera fruit samples were procured from the market locally situated near the Central Institute of Technology Kokrajhar, Assam. Selected fruit, free from damage and spoilage, were chosen and cleansed with running tap water. Seeds from the fruit were separated and sliced into small pieces to facilitate the grinding process. The fruit flesh was then broken up using a mechanical grinder. To remove coarse particles and impurities, the juice was filtered using a sterilized two-layer muslin cloth and centrifuged (C-24L; Remi Cooling Centrifuge, Mumbai, India) [14]. Fruit juices were labeled according to the method of processing as follows: FBTJ (no treatment), PBTJ (pasteurization), and TSBTJ (thermosonication).

### 2.2. Pasteurization of the Juice

Pasteurization was conducted for 60 s at 90 °C in a lab-sized hot water bath (Lab Tech, Daihan Lab Tech India Private Limited, Haryana, India) [18].

### 2.3. Thermosonication of the Juice

Using a 500 W ultrasonic bath, bor-thekera juices were thermosonicated (AnTech, GT Sonic, Qingdao, China). For each experiment, 100 mL of juice was used. Thermosonication was performed using two frequencies (33 kHz and 44 kHz) at three different temperatures (30 °C, 40 °C, and 50 °C) for three different treatment periods (30, 45, and 60 min) [9]. The sonicator bath was equipped with a thermostatic control system. This system continuously monitored the temperature of the liquid in the bath and adjusted the heating element accordingly to maintain the desired set temperature. To avoid light interference, the thermosonication procedure was performed in dark conditions. Juice samples that were thermosonically processed were kept in sterilized, sealed glass bottles. Before the analysis, the juice containers were maintained at a temperature of 4 ± 1 °C [19].

### 2.4. Storage Study

The juice samples thermosonicated at 40 °C at 44 kHz were chosen for the storage study, as the quality retention was the highest. The storage study was conducted in refrigerated conditions (4 ± 1 °C) for one month and the samples were analyzed periodically to determine the changes in the quality parameters of the bor-thekera juice.

### 2.5. Determination of Physicochemical Properties

#### 2.5.1. pH, Total Soluble Solids (TSSs), and Titratable Acidity (TA)

In order to determine the pH of the juice samples, a digital pH meter (EUTECH INSTRUMENTS, ION 2700, pH/mV/°C/°F meter) was used at a temperature of 25 ± 1 °C. The results of every experiment were analyzed in triplicate, and the mean was determined. A digital refractor (Master 3 m ATAGO, Tokyo, Japan) was used to determine the total soluble solids (TSSs) in the bor-thekera juice at an ambient temperature of 25 ± 1 °C. The outcomes were represented in °Brix.

The titratable acidity (TA) of the samples was assessed by titrating a 0.1 N NaOH solution against the juice in the presence of a phenolphthalein indicator (AOAC, 1999). The results were expressed as a citric acid percentage [20].

#### 2.5.2. Cloudiness (CI) and Browning Indexes (BI)

The CI and BI of the samples were assessed by using the procedure given in [21]. Fruit juice samples, both treated and untreated, were spun in a centrifuge (C-24BL; Remi Cooling Centrifuge, India) at a speed of 6000 rpm for ten minutes at ambient temperature. The rest of the supernatant portion was utilized to calculate the CI and BI after the separation of the sample. Utilizing an ultraviolet–visible spectrophotometer (LAMDA-35 Elmer Perkin, Waltham, MA, USA), the absorbance at 660 nm was measured, and the degree of visibility was assessed. The BI was determined by mixing and agitating equal amounts of ethanol, which was the supernatant solution (5 mL each). In an ultraviolet–visible spectrophotometer (LAMDA-35 Elmer Perkin, USA), the BI was identified by analyzing the residue at 420 nm.

### 2.6. Determination of Functional Properties

#### 2.6.1. Total Phenolic Content (TPC) and Total Flavonoid Content (TFC)

The TPC of PBTJ, TSBTJ, and raw juice was assessed using the Folin–Ciocalteu method [5]. The results of the tests were expressed in milligrams of gallic acid equivalent (GAE) per 100 mL of juice. A total of 0.1 mL of fruit juice samples was mixed well with 0.1 mL of Folin–Ciocalteu reagent (which was diluted three times with distilled water). After 3 min, 0.3 mL of 0.19 M sodium carbonate suspension was added, and the resulting solution was left for 2 h prior to its absorbance being measured at 760 nm. The measurements of the calibration curve employed gallic acid as its standard.

The TFC of the bor-thekera juice samples was calculated using the spectrophotometric technique [22]. The TFC levels were displayed in milligrams of quercetin equivalent (QE) per milliliter of fruit juice sample. Six-milliliter juice samples were collected, as well as a 0.2 mL solution containing 0.42 M aluminum chloride and 0.178 M sodium potassium tartrate. Then, 5.6 mL of water from distillation was added to the above mixture and thoroughly mixed. The entire mixture was allowed to incubate in the dark for 30 min before reading its absorbance at 415 nm with a spectrophotometer (LAMDA-35 Elmer Perkin, USA). Catechin served as the measurement curve’s standard.

#### 2.6.2. Antioxidant Activity (AO)

The AO of the bor-thekera juice samples was examined using the DPPH free radical scavenging method [18]. An identified aliquot (2 mL) of the fruit juice samples was obtained. The specimen was treated with 2 mL of DPPH solution (0.0002 M in ethanol) and set aside for 30 min at ambient temperature (25 ± 1 °C). An identical process was followed for the blank, but ethanol was employed instead of the specimen. The absorbance levels of the specimens were determined at 517 nm with an ultraviolet (UV) spectrophotometer (LAMDA-35 Elmer Perkin, USA).

#### 2.6.3. Ascorbic Acid Content (AAC)

The titration (iodine) method was used to estimate the AA content of each juice sample (AOAC, 2000). The AA content of the bor-thekera juices was expressed as mg/100 mL of juice sample [20]. In a 100 mL cylindrical flask, 0.25 g of starch was dissolved and 50 mL of close-to-boiling water was added. Before use, the mixture was stirred to combine and allowed to cool. In a 100 mL beaker, 2 g of potassium iodide and 1.3 g of iodine were added. A few milliliters of water that had been distilled were added and whirled for a few minutes until the iodine was dissolved. The solution of iodine was moved to a 1 L volumetric flask, and filtered water was used to rinse out any remaining solution. Distilled water was added until the solution reached 1 L. A 20 mL portion of the specimen solution (treated/untreated juice) was transferred to a 250 mL cylindrical flask, along with 150 mL of distilled water and 1 mL of the starch indicator solution. The specimen was adjusted using a 0.005 M solution of iodine. The process of titration ended with the first enduring trace of a dark blue-black color caused by the starch–iodine convolution.

### 2.7. Microbial Analysis

Each sample was subjected to a total viable count (TVC) and yeast and mold count (YM) to determine the microbiological activity. The total plate counts were calculated using the spread plating method. After the juice samples were added to the plate count agar medium, the plates were allowed to incubate at 37 ± 1 °C for 48 h.

The juice samples were placed on potato dextrose agar plates and incubated at 25 ± 1 °C for 120 h to determine the presence of yeasts and molds. The results of the microbiological analysis were expressed as log CFU/mL of juice [7].

### 2.8. ANN Experimental Modeling

The ANN was developed using the MATLAB Neural Network Toolbox, which provides a multimedia platform for mathematical computing, visualization, and coding. It was composed of a hidden layer between the input and output layers, as well as modules that use an error backpropagation algorithm to derive the prediction mistake based on the weight. A feed-forward neural network (FFNN) was developed using the Levenberg–Marquardt (LM) combined backpropagation (BP) algorithm [5]. The Levenberg–Marquardt algorithm used 15% of this data for testing, 15% for validity transaction information, and 70% for training. The input variables and their limits were determined according to initial research conducted in laboratories. Storage time X1 (0–30 days), treatment temperature X2 (40 °C), and treatment time X3 (30–60 min) were selected.

The experimental temperature ranged from 30 to 50 °C during the preliminary studies. By involving numerous studies that established the use of an ANN for the processes of various fruit juice extractions [23,24], the maximum value of the treatment temperature range was chosen. As the dependent parameters for choosing the ranges of the independent variables, the effects of storage time, treatment time, and treatment temperature on various responses, including pH, TSSs, CI, TA, BI, TPC, AO, TFC, TVC, AA, and YM, were established. The plot regression function was used to gauge the system’s performance. The generated neural network had three inputs, ten hidden layers, and one output layer. The ANN model with PURELIN was used [25]. PURELIN is commonly used in the output layer of regression equations, where the objective is to anticipate an upcoming value. Because linear activation imposes no limitations or non-linear changes on the output, the neural network can predict any actual value output. To assess the efficacy of the artificial neural network (ANN) models, Equations (1)–(3) were used to calculate the determination coefficient (R^2^), root-mean-square error (RMSE), and absolute average deviation (AAD).
(1)$$R^{2}=\frac{\sum_{i=1}^{n}\left(x_{experimented}-\bar{x})(y_{predicted}-\bar{y}\right)}{\sqrt{\sum_{i=1}^{n}{\left(x_{experimented}-\bar{x}\right)}^{2}}\;\sqrt{\sum_{i=1}^{n}{\left(y_{predicted}-\bar{y}\right)}^{2}}}.$$
(2)$$MAE=\frac{1}{n}\sum_{i=1}^{n}\left(x_{experimented}-y_{predicted}\right)$$
(3)$$MSE=\frac{1}{n}\sum_{i=1}^{n}(x_{experimented}-y_{predicted}{)}^{2}$$

In this scenario, the variables include *x_i* (representing the experimental data), *y_i* (indicating predicted values), *n* (denoting the total number of observations), and *x* (representing the mean value of the experimental data).

#### Genetic Algorithm (GA) Optimization

The genetic algorithm (GA) arsenal of MATLAB (version R2022a, developed by Math Works Inc., Portola Valley, CA, USA) was used to optimize the neural network. The main variables selected for the GA optimization were the storage time (0–30 days), temperature (40 °C), and sonication duration (30, 45, 60 min). Furthermore, for possible populace, rank, blackjack function, dispersed, and adaptable possible outcomes, the development function, fitness in order expanding function, preference function, overlap function, and alteration function were chosen, respectively. The fitness function (*f*), which is displayed below, was created to maximize each output response [25].
*f* = − (*Y*1 + *Y*2 + *Y*3)(4)
where *Y*1, *Y*2, and *Y*3 are the actual response values that the ANN predicted; the negative sign indicates that the function (*f*) in GA was maximized.

### 2.9. Statistical Analysis

All experiments were conducted in triplicate and results were given as the mean ± standard deviation. The data from the analysis were evaluated by MS Excel (Version 2021) and SPPS 16.0 software (SPSS Inc., Chicago, IL, USA). One-way ANOVA was used for the analysis of data, with *p* < 0.05 indicating a statistically significant value.

## 3. Results and Discussion

The impacts of pasteurization and thermosonication on the physiochemical properties of bor-thekera juice are given below.

### 3.1. pH

Table 1 and Table 2 show the physiochemical parameters of bor-thekera juices treated under thermal and non-thermal treatments. On day 0, slight variations in pH from 2.93 ± 0.04 to 3.13 ± 0.03 for 33 kHz and 2.95 ± 0.02 to 3.13 ± 0.04 for 44 kHz were observed for the TSBTJ. The pH values of the FBTJ and PBTJ samples were observed to be 2.91 ± 0.05 and 3.15 ± 0.03, respectively, on day 0. The slight rise in pH during the thermosonication could have been caused by higher acidic compounds that appeared with the higher frequency. For higher time–temperature combinations, the values were gradually raised. The pH values of all samples (FBTJ, PBTJ, and TSBTJ) increased as the storage time progressed. After 30 days of storage, the pHs of the raw, pasteurized, and thermosonicated juices were observed to be 3.35 ± 0.02, 3.42 ± 0.03, and 3.41 ± 0.02, respectively. The changes in the pH values of the samples may have been due to the formation of new chemical components in the bor-thekera juices, which might have participated in triggering various chemical processes at aggressive ultrasound frequencies [20,26].

### 3.2. TSSs

The TSSs of both samples (FBTJ and PBTJ) were obtained as 6.13 ± 0.13 °Brix (day 0), as shown in Table 1 and Table 2; it has to be noted that the TSSs of FBTJ and FBTJ samples did not differ. The TSS values of the TSBTJ samples were in the range of 6.24 ± 0.17 to 7.06 ± 0.19 °Brix (Table 1) for the juice samples sonicated at 33 kHz, and the samples that were processed at 44 kHz exhibited TSS values of 6.30 ± 0.17 to 7.10 ± 0.17, and 6.13 ± 0.13 °Brix (Table 2). The data revealed that the increase in the TSS values was higher in the juices treated at 44 kHz compared with the treatments at 33 kHz. The level of soluble solids increasing in TSBTJ samples may have been due to the enhanced extraction ability of thermosonication treatments [26].

During the storage study, the TSSs of all samples (treated and untreated) increased gradually. At the end of the storage period, the TSSs of the FBTJ and PBTJ were obtained as 7.50 ± 0.12 and 7.58 ± 0.14 °Brix, respectively. The TSS values of the TSBTJ samples varied between 7.77 ± 0.09 and 8.41 ± 0.14 °Brix on day 30. From the data, it can be understood that the TSSs of the TSBTJ samples were higher compared with those of the FBTJ and PBTJ. The increase in TSSs of the TSBTJ samples may have been due to the development of cavitations, which caused cell damage and led to the creation of tiny channels promoting dehydration, dissolving more solids [27]. The lowered fruit metabolism owing to the decrease in the degradation of organic acids along with cell wall constituents resulted in increased TSS values at the higher ultrasound frequency [14].

### 3.3. TA

The titratable acidity values were recorded as 0.548 ± 0.016% and 0.540 ± 0.017% for the FBTJ and PBTJ samples (Table 1 and Table 2). The TSBTJ samples treated at 33 kHz exhibited TA values between 0.455 ± 0.011 and 0.533 ± 0.011% (Table 1), whereas the bor-thekera juices processed at 44 kHz exhibited TA values between 0.470 ± 0.011 and 0.535 ± 0.18% (Table 2) on day 0. Thermosonication caused a decrease in the titratable acidity of the bor-thekera juices compared with the FBTJ and PBTJ samples, and it has to be noted that the drop in the TA values was greater for higher frequency–time–temperature conditions. The decrease in TA could be explained by the lower energy levels of thermosonication, which might not have been sufficient to alter the molecular structure of some larger molecules [14,27,28].

The observed values for the 30 days of storage were displayed in Table 3, which demonstrated a drop during the storage trial. The readings for TA decreased to 0.422 ± 0.012% for the FBTJ and 0.419 ± 0.013% for the pasteurized juice. For the thermosonicated juice, it ranged from 0.399 ± 0.010 to 0.407 ± 0.009%. The reduction in the titratable acidity values of the juices may be attributed to the use of organic acids throughout fruit metabolism [14]. The decline in the TA of the TSBTJ may have been due to the acidic hydrolysis of polysaccharide compounds, where acid was needed to transform the sugars from a nonreducing form into a reducing form [29].

### 3.4. The Impacts of Pasteurization and Thermosonication on Cloudiness of Bor-Thekera Juice

As observed from Table 1 and Table 2, the CIs of the TSBTJ and PBTJ samples were found to be lower than that of the FBTJ on day 0. The CI of the FBTJ and PSBTJ samples was obtained as 0.0357 ± 0.0013 and 0.0290 ± 0.0019, respectively. Thermosonication treatments reduced the CI of the bor-thekera juices and the CIs were recorded in the ranges of 0.0308 ± 0.0016 to 0.0242 ± 0.0011 and 0.0296 ± 0.0017 to 0.0218 ± 0.0017, respectively, for 33 kHz and 44 kHz (Table 1 and Table 2). The experimental data indicate that the thermosonication at the higher frequency was involved in a higher reduction of the CI of the bor-thekera juice. The improved dispersion of particles (cellulose, hemicelluloses, pectin, etc.) and other minor components present in the bor-thekera juice may have been the cause of the decrease in cloudiness during thermosonication [27]. It was also reported that the disintegration of linear pectin molecules due to thermosonication, which resulted in the reduction in molecular weight, may have also decreased the CI due to weaker network creation [29].

The CIs of all the bor-thekera juice samples were decreased during the 30 days of storage, as presented in Table 3. The CIs of the FBTJ and PBTJ were reduced from the initial values of 0.0357 ± 0.0013 and 0.0290 ± 0.0019 to 0.0198 ± 0.0006 and 0.0187 ± 0.0016 at the end of the storage period (day 30). For the TSBTJ juices, the CI was obtained in the range of 0.171 ± 0.0005 to 0.0147 ± 0.0004 at day 30. Juices that were thermosonicated produced a lesser cloud value due to the breakdown of various molecules present in the juice samples. The level of dissolved particles in the dispersed phase was correlated with the transparency of the bor-thekera juice. Juice becomes clear if the particles are sufficiently distributed in the dispersion medium [17,30].

### 3.5. The Impacts of Pasteurization and Thermosonication on Browning Index (BI) of Bor-Thekera Juice

The influence of processing on the browning index of the bor-thekera juices is presented in Table 1 and Table 2, and the BI values of the FBTJ and PBTJ were obtained as 0.0318 ± 0.010 and 0.0361 ± 0.014, respectively. The BI values of thermosonicated juices at 33 kHz and 44 kHz were registered in the ranges of 0.0479 ± 0.011–0.0598 ± 0.014 and 0.0480 ± 0.012–0.0594 ± 0.014, respectively. From the data, the thermal treatment did not cause higher changes in the BI compared with the TSBTJ samples. In the thermosonicated samples, the values were found to be elevated as the treatment time, temperature, and frequency increased.

The alteration in the browning index of TSBTJ samples could be correlated with the oxidation of AA and the TPC [8]. The increase in the BI of the TSBTJ seems to have been caused by the AA breakdown. Carbonyl radicals, which are produced when ascorbic acid oxidation occurs, have the ability to catalyze non-enzymatic processes. Furthermore, the formation of furfural during the disintegration process could raise the BI of TSBTJ [14].

During the storage study, all bor-thekera juices (treated and untreated) showed a substantial rise in the observed values of browning with increasing time (Table 3). On the 30th day (after the full storage period), the BI values of the FBTJ, PBTJ, and TSBTJ samples were obtained as 0.0437 ± 0.013, 0.0492 ± 0.007, and 0.0618 ± 0.009 to 0.0649 ± 0.014, respectively. The juice samples that were thermosonically treated showed the greatest increase. A rise in the BI was seen in all samples as the period of storage proceeded. The rise in the BI might be correlated with non-enzymatic interactions between sugars and amino acids that eventuate during the storage period [31].

### 3.6. The Impacts of Pasteurization and Thermosonication on the TPC and TFC of Bor-Thekera Juice

From Figure 1, it can be observed that the TPC of the bor-thekera juice after pasteurization increased compared with the TPC of the FBTJ, and the values were found to be 380.83 ± 14.09 and 289.23 ± 11.64 mg GAE/100 mL, respectively, for the PBTJ and FBTJ. The TPC values of the bor-thekera juices were also elevated by thermosonication treatments; the values were measured at 33 kHz and 44 kHz, respectively, and ranged from 303.5 ± 10.08 to 551 ± 14.83 mg GAE/100 mL and 330.66 ± 13.60 to 588.16 ± 11.17 mg GAE/100 mL. It was documented that the increase in the TPC of the TSBTJ may be ascribed to the release of phenolic compounds from the cell walls of bor-thekera fruit through cavitations created by TS [8]. In our study, it was noticed that the higher frequency and longer time resulted in a higher phenolic content. Treatments at 44 kHz yielded higher TPC values compared with the processing of bor-thekera juices at 33 kHz. This rise might be explained by the release of bound molecular oxygen [32].

The impact of pasteurization and thermosonication on the TFC of the bor-thekera juice is depicted in Figure 2. The highest value of TFC was obtained when thermosonication was carried out at 40 °C using 44 kHz for 60 min. The TFCs of the FBTJ and PBTJ were found to be 45.86 ± 1.79 QE/100 mL and 57.37 ± 2.54 QE/100 mL, respectively. The TFC values were increased due to the application of thermosonication and the values were noted in the ranges of 50.42 ± 2.31 to 96.07 ± 4.18 QE/100 mL and 53.7 to 96.76 QE/100 mL for the treatments at 33 kHz and 44 kHz, respectively. Higher TFCs were obtained due to the exposure of bor-thekera juices to higher frequencies, which resulted in increased cavitation. Consequently, numerous cell walls were disrupted, which led to the discharge of flavonoid components. The attachment of OH^−^ compounds to the aromatic ring of phenolic compounds because of thermosonication could contribute to the increase in the total phenolic and flavonoid concentrations [22] in the bor-thekera juices.

Table 3 shows the effects of thermosonication and pasteurization on the TPC and TFC of the bor-thekera juices. As observed from this table, all samples displayed a decline in the TFC and TPC levels of the bor-thekera juices during refrigerated storage. On day 30, the TPCs of the FBTJ, PBTJ, and TSBTJ were found to be 190.13 ± 16.02 mg GAE/100 mL, 197.98 ± 16.32 mg GAE/100 mL, and 326.27 ± 15.66 to 592.21 ± 16.60 mg GAE/100 mL, respectively. The results for the TFCs of the FBTJ, PBTJ, and TSBTJ were recorded as 22.68 ± 1.07 QE/100 mL, 28.63 ± 1.21 QE/100 mL, and 45.43 ± 1.58–49.31 ± 2.04 QE/100 mL (Table 3), respectively. Similar results were documented earlier for different juices by researchers [23,26]. The decrease in the TPC and TFC of the TSBTJ during storage may have been caused by the generation of hydroxyl radicals. These chemicals might help to hydrolyze phenolic glycosidic moieties to produce phenolic compounds with poorer stability [33]. Better retention of the TPC and TFC in the TSBTJ sample might have been interrelated with the cavitation effect of thermosonication, which excluded dissolved oxygen and a slowing down of the oxidation of phenolic substances during storage [31].

### 3.7. The Impacts of Pasteurization and Thermosonication on the Antioxidant Activity of Bor-Thekera Juice

The changes in the antioxidant activities of the bor-thekera juices after processing are depicted in Figure 3. The antioxidant activities of the FBTJ and PBTJ were noted as 60.40 ± 2.33% and 59.90 ± 2.08%, respectively. Thermosonication increased the antioxidant activities of the bor-thekera juices and the highest antioxidant activities were registered for the juices treated at 40 °C using 44 kHz for 45 min. The antioxidant activities of the TSBTJ samples treated at 33 kHz and 44 kHz were found to be in the ranges of 67.60 ± 2.63% to 81.27 ± 3.29% and 71.09 ± 2.65% to 81.86 ± 3.07%, respectively. The increase in the antioxidant activity of the TSBTJ compared with the FBTJ and PBTJ may be related to the high retention of phenolic compounds, as well as the high extraction rates of antioxidant components (phenol and ascorbic acid) due to cavitation during the thermosonication treatments [16,29,34]. It was also reported that the high AO of TSBTJ samples may be linked with the discharge of bound phenolic compounds after thermosonication [17].

As observed in Table 3, the antioxidant activities of the bor-thekera juices during storage followed a decreasing trend irrespective of the treatments. But the antioxidant activities of the TSBTJ samples always remained higher compared with the FBTJ and PBTJ samples. The antioxidant activities on day 30 for the FBTJ, PBTJ, and TSBTJ were observed to be 26.18 ± 1.42%, 26.18 ± 1.71%, and 42.17 ± 3.56% to 51.17 ± 2.29%, respectively. It was hypothesized that the released oxygen due to thermosonication could positively affect the antioxidant activities of the TSBTJ samples by improving their stability [35]. But the possible explanation for the reduction in the TSBTJ samples could be related to the creation of hydroxyl radicals, which might have been involved in the deterioration of antioxidants, such as phenolic compounds [35].

### 3.8. The Impacts of Pasteurization and Thermosonication on the Ascorbic Acid Content (AAC) of Bor-Thekera Juice

The effects of different processing methods on the ascorbic acid contents of bor-thekera juices are presented in Figure 4. From this figure, it can be inferred that the AAC of the PBTJ and TSBTJ samples were found to be decreased compared with the FBTJ. Pasteurization showed the least retention of ascorbic acid (AA), and the values for the FBTJ and PBTJ were noted as 47.85 ± 1.39 and 33.19 ± 1.15 mg/100 mL, respectively. As shown in Figure 4, the highest retention of AA was observed for the lower temperature employed for the thermosonication treatments. When the treatment time and frequency increased along with the temperature, the decline in AA was also increased during thermosonication. The AAC for the TSBTJ was found to vary between 36.24 ± 1.25 to 47.75 ± 1.83 mg/100 mL and 37.12 ± 1.17 to 46.34 ± 2.09 mg/100 mL, respectively, for 33 kHz and 44 kHz. According to earlier reports on vitamin C oxidation, AA can be reduced by aerobic and anaerobic processes when exposed to light, high temperatures, and pressure. By generating free radicals, sonication may also reduce the AA level in the juice [36].

The impact of storage on the AAC is presented in Table 3. As seen from this Table 3, the AAC of all samples followed a decreasing trend over the 30-day storage period. The AAC for the FBTJ and PBTJ samples were recorded as 21.88 ± 0.92 mg/100 mL and 15.39 ± 0.76 mg/100 mL, respectively, at the end of the storage period (day 30). The bor-thekera juices processed under thermosonication (44 kHz, 40 °C) showed values in the range of 19.14 ± 0.79–24.31 ± 0.88 mg/100 mL. The reduction in the AAC was primarily associated with oxidation reactions that occurred in the bor-thekera juice during storage; more specifically, oxygen that persisted in the bor-thekera juices at the initial period of storage was involved with the aerobic oxidation, along with the later decline in the ascorbic acid levels involved with the anaerobic degradation when the oxygen levels were exhausted [31]. At the same time, the reduction in the TSBTJ during storage was comparatively lower due to the reduction of dissolved oxygen levels and enzyme inactivation through thermosonication [37].

### 3.9. The Impacts of Pasteurization and Thermosonication on Microbial Populations of Bor-Thekera Juice

#### Total Viable Count (TVC) and Yeast and Mold Count (YMC)

The experimental data showed that the processing methods had an impact on the microbial populations (total viable count (TVC) and yeast and mold count (YMC)) in the bor-thekera juices, as given in Table 4. The TVC and YMC on day 0 were determined to be 4.95 ± 0.03 log CFU/mL and 4.79 ± 0.04 log CFU/mL for the FBTJ samples. The PBTJ samples did not show any microbial growth. The TVC of the TSBTJ had lower microbial activity ranging from 3.92 ± 0.03 log CFU/mL to 3.81 ± 0.03 log CFU/mL at 33 kHz and 3.88 ± 0.02 log CFU/mL to 3.70 ± 0.03 log CFU/mL at 44 kHz, whereas the YMC of the TSBTJ had lower activity ranging from 3.75 ± 0.03 log CFU/mL to 3.67 ± 0.02 log CFU/mL at 33 kHz and 3.72 ± 0.04 log CFU/mL to 3.44 ± 0.04 log CFU/mL at 44 kHz. A decrease in the microbial counts was seen after the thermosonication treatments on the bor-thekera juices compared with the FBTJ. It must be observed that as the implementation time and temperature were increased, the rate of microbial population decline accelerated. From previous works, it was found that the greater vulnerability of microbes to heat was what caused thermosonication’s effect of reducing microbial populations, along with the pressure and low pH as a consequence of cavitation and modification in the cell structure of the microbes [28].

During the storage period, a gradual increase in the microbial populations was noticed for the bor-thekera juices (Table 4). The TVC and YMC of the FBTJ were found to be in the ranges of 4.95 ± 0.03 log CFU/mL to 6.04 ± 0.04 log CFU/mL and 4.79 ± 0.04 log CFU/mL to 6.14 ± 0.03 log CFU/mL, respectively. The PBTJ samples did not produce any detectable growth initially for 3 days. After this period, the microbial counts of the PBTJ samples were increased and the values for the TVC and YMC were recorded in the ranges of 2.25 ± 0.02 log CFU/mL to 3.70 ± 0.03 log CFU/mL and 2.53 ± 0.02 log CFU/mL to 3.68 ± 0.03 log CFU/mL, respectively. The microbial counts of the TSBTJ samples were found to be lower compared with the FBTJ and higher compared with the PBTJ during the storage study. The TPCs of the TSBTJ samples were found to be in the range of 3.86 ± 0.03 log CFU/mL to 4.94 ± 0.04 log CFU/mL, whereas YMC values were found to be between 3.67 ± 0.03 log CFU/mL and 5.03 ± 0.03 log CFU/mL. Though the pasteurization was the most effective, thermosonication also showed good preservation of the juices. The inactivation of the microbes during thermosonication may have been caused by the effects of the cavitation on the inner membrane of microbes, including the rapid rise in the liquid temperatures and pressures, as well as the production of hydroxyl radicals. Along with cavitation, the internal structure is also altered by physical phenomena, such as shearing forces and turbulence [38]. The sequential increase in the microbial counts of TSBTJ during the storage period can be correlated with the recovery of injured microbial cells and the growth of remaining cells [31].

### 3.10. GA Optimization and Modeling of Thermosonicated Bor-Thekera Juice Using ANN

The experimental data, which consisted of 30 data points, was first divided into three sets at random for training, validation, and testing, comprising roughly 75%, 15%, and 15% of the total datasets. Training establishes the network parameters, validation determines how robust the network parameters are, and testing limits the error in the network parameters. Table 5 presents the experimental and anticipated values [9,39]. The neural network framework employed in the present research had only one input, layers that were concealed, and output layers, as shown in Figure 5. To create the most effective ANN model for anticipating responses, the number of neurons in the layer that is hidden must be decided. This was confirmed by repeatedly training the network after the lowest MSE and highest R were reached. Following several training sessions, the hidden layer’s number of neurons was set to 10 because it had minimum MSEs of 0.386, 0.370, 0.118, 0.688, 0.487, 0.188, 0.141, 0.805, 0.591, 0.169, and 0.587 for each response (dependent parameters), and maximum Rs of 0.998, 0.994, 0.997, and 0.997 for the training, testing, validation, and all datasets, respectively. Figure 6, Figure 7 and Figure 8 show the ANN configuration’s post-training performance, error histogram, and regression analysis. Figure 9 depicts the performance of MSE as the epoch number increased during training, validation, and testing. It demonstrates that as the number of epochs increased, the system showed high performance, with the best approval efficiency obtained at 12 epochs and an MSE corresponding to 197.4046. According to the error histogram in Figure 7, with the most data points possible, the error was 0.2767, which is a value that is near 0. The results of the regression analysis shown in Figure 8 demonstrate the accuracy of the expected outcomes (output) for training, evaluation, and validation, as well as the regression model fit with the actual outcomes (target). Table 5 shows the anticipated outcomes (the outputs) of the responses found using the ANN model. All four regression lines had R values close to one. Furthermore, the similarity between the actual (target) and predicted (output) values in Table 5 verified the fit’s reliability. The ideal conditions for the extraction of the bor-thekera juice when employing thermosonication were obtained in 12 iterations using ANN and were as follows: 30 min at 40 °C. As the most suitable procedure variables were implemented in the real-life extraction of bor-thekera juice, the responses were determined through experimentation and found to be well aligned with the predicted values, demonstrating that the generated model of ANN had good prediction and optimization ability. Contrary to initial concerns about the feasibility of a 30 min duration in an industrial context, the analysis considered the advantages gained in terms of improved product quality, reduced energy consumption, and enhanced safety measures. Additionally, advancements in industrial equipment and processes have made it increasingly practical to implement slightly longer processing times for certain applications. Thus, the findings suggest that a 30 min processing time is not only viable but could also yield substantial benefits in terms of overall process efficiency and product quality in an industrial setting.

## 4. Conclusions

This research work was performed to assess the impact of different thermosonication treatments on quality attributes, such as functional, physicochemical, and microbial populations, of bor-thekera juice that was placed in refrigerated storage for 30 days. From the research data, it can be understood that the TSBTJ samples showed improvements in the TPC, TFC, and antioxidant activities and a lower loss in ascorbic acid compared with the FBTJ, and was superior to the PBTJ in all functional properties on day 0. The microbial inactivation was better in the PBTJ compared with the TSBTJ samples, but the microbial levels were lower than in the FBTJ. It was observed that the TSBTJ treated at 40 °C and 44 kHz exhibited better quality aspects compared with other thermosonication treatment temperatures. During the storage, the values of functional attributes (AA, AO, TPC, TFC) decreased gradually in all samples irrespective of the treatments. But the TSBTJ samples displayed greater retainment of functional components compared with the FBTJ and PSBTJ. The microbial stability of the TSBTJ was also observed to be good during the storage period compared with the FBTJ. Three neurons were used in the input layer, ten in the single hidden layer, and eleven in the result layer to create an ideal type of ANN model. Following training, the network produced minimum MSE values of 0.386, 0.370, 0.118, 0.688, 0.487, 0.188, 0.141, 0.805, 0.591, 0.169, and 0.587, and maximum R values of 0.998, 0.994, 0.997, and 0.997 for the training, testing, validation, and all datasets, respectively. This suggests both a good generalization of the ANN and an excellent fit between the actual and predicted values. For the highest desirable values of responses, the following extraction parameters were optimized using the ANN: incubation time of 30 min and incubation temperature of 40 °C at 44 kHz. The measured values of responses were compared with the predicted values of outcomes in the optimized conditions generated by the ANN model and they showed good agreement. In light of this, it can be said that the developed ANN model is a useful quantitative tool for optimizing process variables for enhancing the quality attributes of bor-thekera juice. From the research work, it can be concluded that thermosonication may be effective for the processing of bor-thekera juice with greater levels of functional attributes and reduced microbial activity.

## Figures and Tables

**Figure 1 foods-13-00497-f001:**
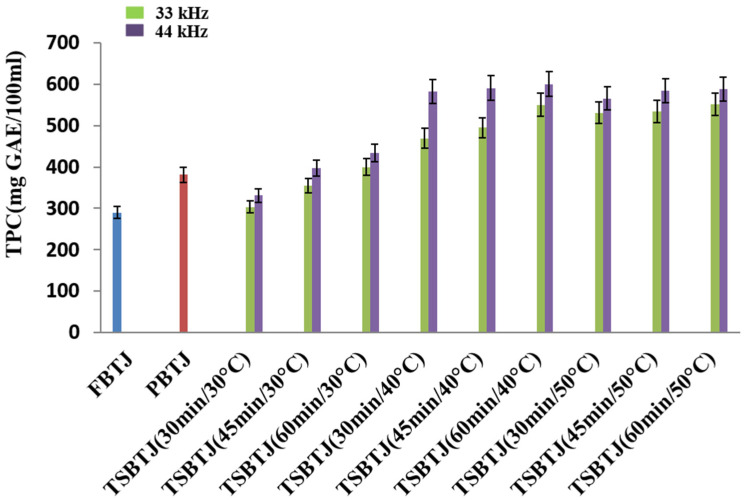
The effects of pasteurization and thermosonication (33 kHz and 44 kHz) on total phenolic content of bor-thekera juice. FBTJ: Fresh bor-thekera juice; PBTJ: Pasteurized bor-thekera juice.

**Figure 2 foods-13-00497-f002:**
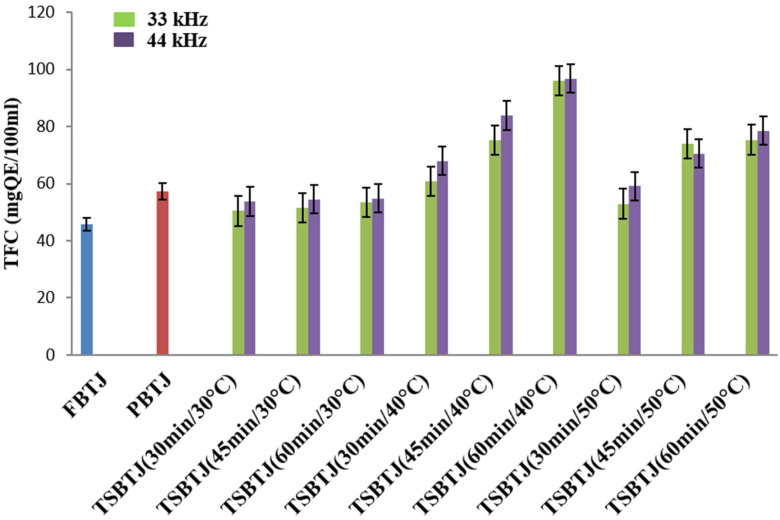
The effect of pasteurization and thermosonication (33 kHz and 44 kHz) on total flavonoid content of bor-thekera juice. FBTJ: Fresh bor-thekera juice; PBTJ: Pasteurized bor-thekera juice.

**Figure 3 foods-13-00497-f003:**
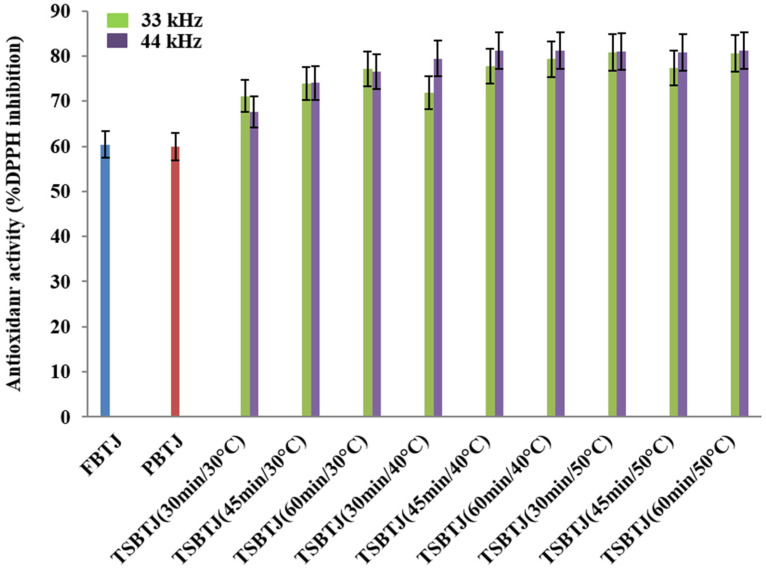
The effects of pasteurization and thermosonication (33 kHz and 44 kHz) on antioxidant activity of bor-thekera juice. FBTJ: Fresh bor-thekera juice; PBTJ: Pasteurized bor-thekera juice.

**Figure 4 foods-13-00497-f004:**
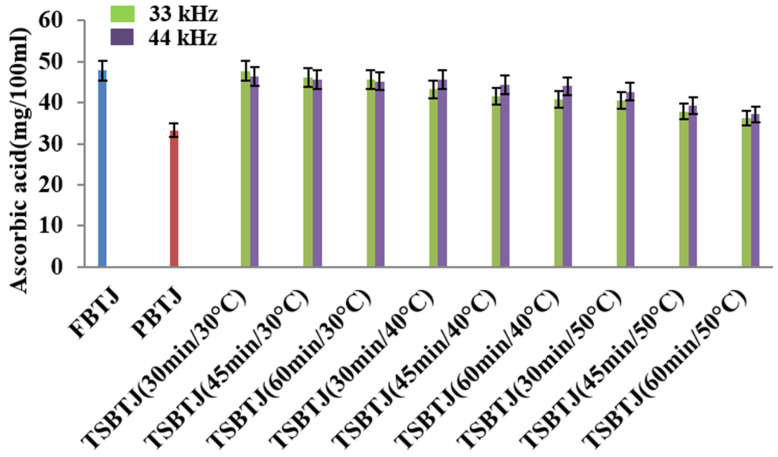
The effect of pasteurization and thermosonication (33 kHz and 44 kHz) on ascorbic acid content of bor-thekera juice. FBTJ: Fresh bor-thekera juice; PBTJ: Pasteurized bor-thekera juice.

**Figure 5 foods-13-00497-f005:**
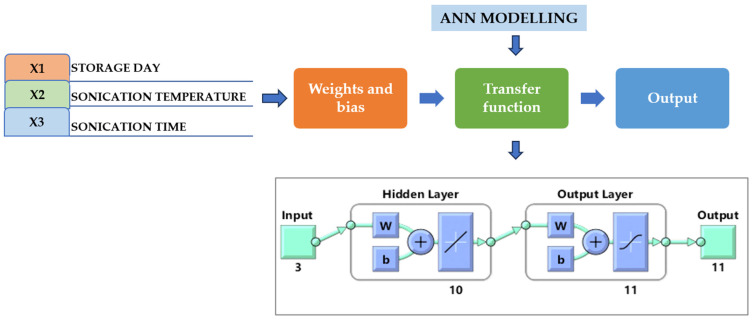
Neural network model with a single input and hidden and output layers. Under the input, 3 indicates the independent parameters (storage days, temperature, and time of TS treatment), hidden layer includes 10 number of neurons, while output 11 indicates the dependent parameters (pH, TSS, TA, CI, BI, AOA, AAC, TPC, TFC, TVC, and YMC). W, weight; b, bias.

**Figure 6 foods-13-00497-f006:**
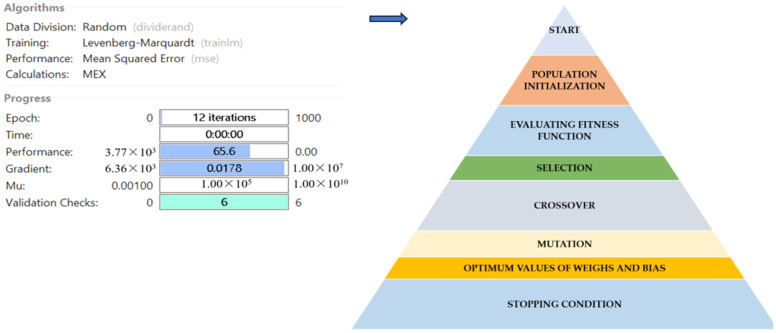
Post-training performance in ANN configuration with GA optimization. Blue arrow indicates the next step after ANN modelling.

**Figure 7 foods-13-00497-f007:**
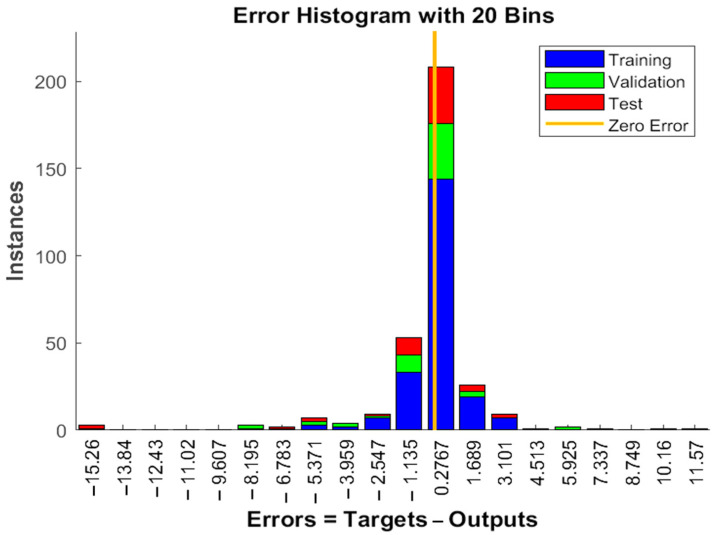
Error histogram in ANN configuration. Error histogram of the developed ANN model. Instances, the number of data points or examples that fall into different error or loss bins within the histogram; zero error, model’s predictions match the true target values exactly for those instances; ANN, artificial neural network.

**Figure 8 foods-13-00497-f008:**
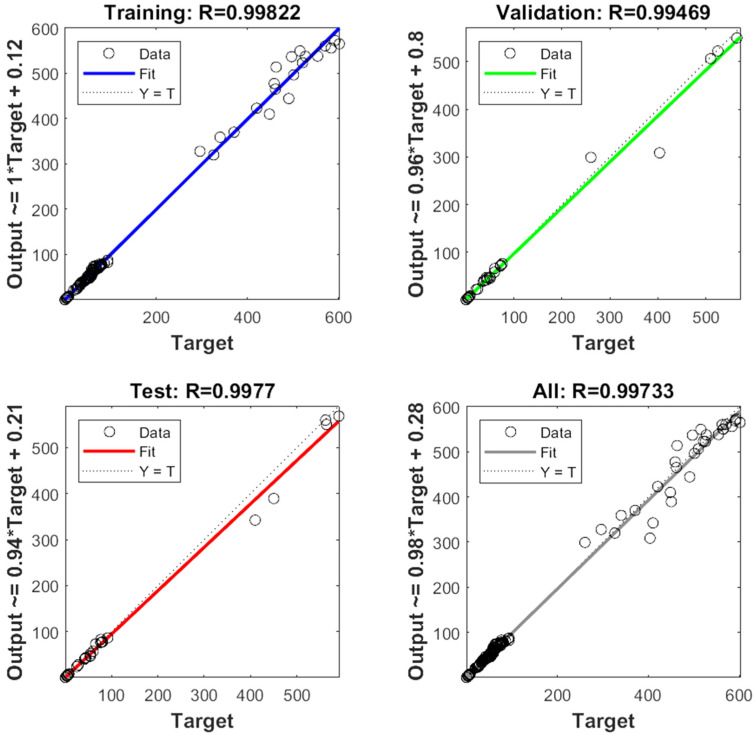
Regression analysis of ANN configuration. The correlation coefficient (R) of training, test, and all data of well-developed ANN model. MSE, mean square error; output, predicted data generated by ANN; target, expected outcome for the given input; training, adjustment in its internal parameters (weights and biases) based on a labeled dataset; testing, accuracy, precision, recall, and mean square error depending on the type of task (classification or regression); validation, monitor and fine-tune the model’s performance, ANN, artificial neural network.

**Figure 9 foods-13-00497-f009:**
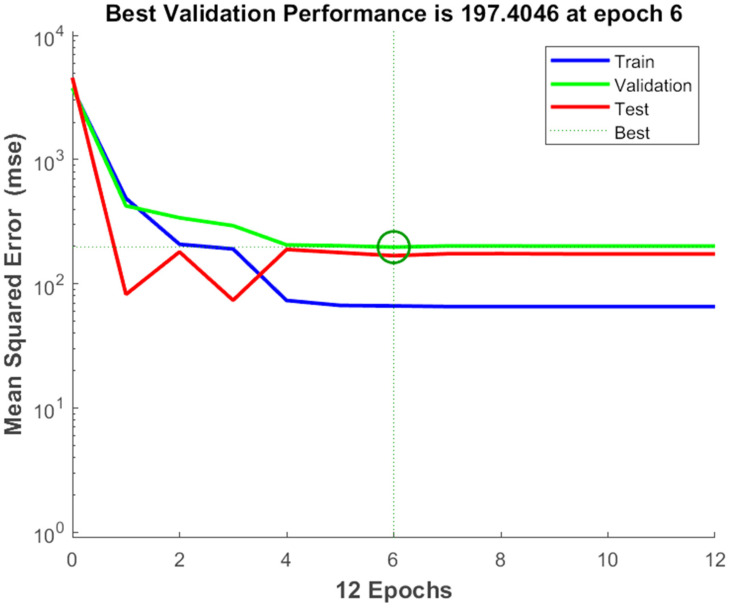
The performance of MSE with respect to the increase in the number of epochs. Epochs: the number of epochs is a hyperparameter that needs to be set by the user and refers to number of time iterations through the entire training dataset. MSE, mean square error. A green circle might indicate the MSE at a particular stage of training, allowing you to observe how the model’s performance evolves over time. Generally, the goal is to see the MSE decrease over time, reflecting improved model performance and better alignment between predicted and actual values.

**Table 1 foods-13-00497-t001:** The effects of pasteurization and thermosonication (33 kHz) on pH, TSSs, TA, cloudiness, and browning index of bor-thekera juice ^1^.

Treatment/Condition	pH	TSSs (°Brix)	TA (%)	Cloudiness	Browning Index
FBTJ	2.91 ± 0.05 ^a^	6.13 ± 0.13 ^a^	0.548 ± 0.016 ^a^	0.0357 ± 0.0013 ^a^	0.0318 ± 0.010 ^a^
PBTJ	3.15 ± 0.03 ^b^	6.13 ± 0.13 ^a^	0.540 ± 0.017 ^ab^	0.0290 ± 0.0019 ^bcd^	0.0361 ± 0.014 ^a^
TSBTJ (30 min/30 °C)	2.93 ± 0.04 ^ac^	6.24 ± 0.17 ^a^	0.533 ± 0.011 ^abc^	0.0308 ± 0.0016 ^b^	0.0479 ± 0.011 ^a^
TSBTJ (45 min/30 °C)	3.04 ± 0.02 ^adc^	6.27 ± 0.18 ^a^	0.520 ± 0.017 ^abcd^	0.0265 ± 0.0013 ^cde^	0.0543 ± 0.014 ^a^
TSBTJ (60 min/30 °C)	3.12 ± 0.06 ^bd^	6.39 ± 0.16 ^ab^	0.514 ± 0.012 ^abcd^	0.0231 ± 0.0018 ^e^	0.0583 ± 0.012 ^a^
TSBTJ (30 min/40 °C)	2.94 ± 0.04 ^ac^	6.35 ± 0.19 ^ab^	0.499 ± 0.011 ^bcd^	0.0292 ± 0.0014 ^bcd^	0.0526 ± 0.013 ^a^
TSBTJ (45 min/40 °C)	3.03 ± 0.03 ^dc^	6.87 ± 0.13 ^bc^	0.502 ± 0.013 ^cd^	0.0251 ± 0.0016 ^de^	0.0587 ± 0.013 ^a^
TSBTJ (60 min/40 °C)	3.12 ± 0.02 ^bd^	6.92 ± 0.18 ^c^	0.494 ± 0.014 ^cde^	0.0243 ± 0.0013 ^e^	0.0591 ± 0.012 ^a^
TSBTJ (30 min/50 °C)	2.98 ± 0.05 ^ab^	6.79 ± 0.14 ^c^	0.494 ± 0.012 ^cde^	0.0298 ± 0.0012 ^bc^	0.0542 ± 0.011 ^a^
TSBTJ (45 min/50 °C)	3.10 ± 0.03 ^cd^	7.00 ± 0.16 ^c^	0.484 ± 0.013 ^de^	0.0253 ± 0.0014 ^de^	0.0579 ± 0.016 ^a^
TSBTJ (60 min/50 °C)	3.13 ± 0.03 ^ab^	7.06 ± 0.19 ^c^	0.455 ± 0.011 ^e^	0.0242 ± 0.0011 ^e^	0.0598 ± 0.014 ^a^

^1^ Values with different superscript small letters in the same column (a–e) are significantly different (*p* < 0.05) from each other. Mean values of three replicates. FBTJ—fresh bor-thekera juice; PBTJ—pasteurized bor-thekera juice; TSBTJ—thermosonicated bor-thekera juice; TSSs—total soluble solids; TA—titrable acidity.

**Table 2 foods-13-00497-t002:** The effects of pasteurization and thermosonication (44 kHz) on pH, TSSs, TA, cloudiness, and browning index of bor-thekera juice ^1^.

Treatment/Condition	pH	TSSs (°Brix)	TA (%)	Cloudiness	Browning Index
FBTJ	2.91 ± 0.05 ^d^	6.13 ± 0.13 ^d^	0.548 ± 0.016 ^a^	0.0357 ± 0.0013 ^a^	0.0318 ± 0.010 ^a^
PBTJ	3.15 ± 0.03 ^a^	6.13 ± 0.13 ^d^	0.540 ± 0.017 ^ab^	0.0290 ± 0.0019 ^b^	0.0361 ± 0.014 ^a^
TSBTJ (30 min/30 °C)	2.95 ± 0.02 ^cd^	6.30 ± 0.17 ^cd^	0.535 ± 0.018 ^ab^	0.0296 ± 0.0017 ^b^	0.0480 ± 0.012 ^a^
TSBTJ (45 min/30 °C)	3.05 ± 0.05 ^abc^	6.35 ± 0.14 ^cd^	0.517 ± 0.012 ^abc^	0.0241 ± 0.0016 ^bc^	0.0525 ± 0.013 ^a^
TSBTJ (60 min/30 °C)	3.10 ± 0.04 ^ab^	6.43 ± 0.14 ^cd^	0.512 ± 0.011 ^abcd^	0.0227 ± 0.0015 ^bcd^	0.0589 ± 0.011 ^a^
TSBTJ (30 min/40 °C)	2.92 ± 0.03 ^d^	6.56 ± 0.16 ^bcd^	0.504 ± 0.014 ^bcd^	0.0259 ± 0.0013 ^bcde^	0.0512 ± 0.015 ^a^
TSBTJ (45 min/40 °C)	3.07 ± 0.05 ^ab^	6.73 ± 0.12 ^abc^	0.499 ± 0.016 ^bcd^	0.0222 ± 0.0016 ^cde^	0.0570 ± 0.011 ^a^
TSBTJ (60 min/40 °C)	3.10 ± 0.03 ^ab^	6.89 ± 0.19 ^ab^	0.489 ± 0.012 ^cd^	0.0214 ± 0.0012 ^de^	0.0584 ± 0.010 ^a^
TSBTJ (30 min/50 °C)	3.01 ± 0.02 ^bcd^	7.03 ± 0.16 ^a^	0.484 ± 0.016 ^cd^	0.0273 ± 0.0011 ^de^	0.0568 ± 0.009 ^a^
TSBTJ (45 min/50 °C)	3.09 ± 0.05 ^ab^	7.08 ± 0.14 ^a^	0.481 ± 0.017 ^cd^	0.0257 ± 0.0015 ^de^	0.0583 ± 0.006 ^a^
TSBTJ (60 min/50 °C)	3.13 ± 0.04 ^a^	7.10 ± 0.17 ^a^	0.470 ± 0.011 ^d^	0.0218 ± 0.0017 ^e^	0.0594 ± 0.014 ^a^

^1^ Values with different superscript small letters in the same column (a–e) are significantly different (*p* < 0.05) from each other. Mean values of three replicates. FBTJ—fresh bor-thekera juice; PBTJ—pasteurized bor-thekera juice; TSBTJ—thermosonicated bor-thekera juice; TSSs—total soluble solids; TA—titrable acidity.

**Table 3 foods-13-00497-t003:** Experimental output for the bor-thekera juice thermosonicated at 44 kHz ^1^.

SDs	Time	TT	PH	TSSs	TA	CI	BI	AO	AAC	TPC	TFC	TVC	YM
0	30	40	2.92	6.56	0.49	0.026	0.051	79.48	45.64	583.73	67.96	3.86	3.77
0	45	40	3.07	6.73	0.50	0.022	0.057	81.27	44.32	601.62	93.85	3.73	3.53
0	60	40	3.10	6.89	0.49	0.021	0.058	81.18	43.98	592.21	93.87	3.74	3.46
1	30	40	3.01	6.63	0.47	0.025	0.055	76.19	44.31	564.18	65.48	3.92	4.13
1	45	40	3.09	7.03	0.46	0.022	0.059	78.22	43.21	570.53	85.93	3.79	3.71
1	60	40	3.11	7.09	0.46	0.020	0.060	80.40	42.75	590.94	91.56	3.81	3.70
3	30	40	3.10	6.82	0.46	0.023	0.055	75.79	43.98	554.18	62.31	4.00	4.19
3	45	40	3.13	7.16	0.46	0.021	0.059	75.52	42.79	564.75	75.46	3.94	3.76
3	60	40	3.17	7.20	0.45	0.020	0.061	78.61	41.69	562.42	76.84	3.93	3.77
5	30	40	3.17	7.14	0.46	0.022	0.055	74.26	42.56	521.08	61.21	4.08	4.28
5	45	40	3.19	7.29	0.45	0.020	0.059	74.18	39.78	527.38	75.40	3.94	3.78
5	60	40	3.20	7.34	0.43	0.019	0.062	75.82	39.59	515.38	75.78	3.96	3.81
7	30	40	3.17	7.27	0.46	0.022	0.060	71.85	38.76	510.14	59.47	4.62	4.41
7	45	40	3.21	7.47	0.45	0.020	0.060	72.28	36.15	525.19	59.89	4.02	4.25
7	60	40	3.24	7.56	0.43	0.018	0.062	75.27	35.88	496.95	62.92	3.99	4.22
10	30	40	3.18	7.35	0.45	0.021	0.060	69.92	36.08	458.94	56.01	4.69	4.47
10	45	40	3.24	7.51	0.44	0.019	0.061	68.43	35.78	501.48	57.51	4.12	4.32
10	60	40	3.26	7.61	0.42	0.017	0.063	63.77	32.79	463.13	58.18	4.01	4.29
15	30	40	3.21	7.49	0.44	0.021	0.060	58.36	34.19	420.46	54.92	4.71	4.63
15	45	40	3.25	7.64	0.43	0.019	0.062	63.70	33.27	490.91	56.36	4.31	4.50
15	60	40	3.28	7.69	0.41	0.016	0.064	61.09	29.68	461.52	55.39	4.25	4.44
20	30	40	3.24	7.56	0.43	0.020	0.061	52.67	28.77	370.77	50.29	4.78	4.76
20	45	40	3.28	7.72	0.42	0.018	0.062	60.12	27.91	450.36	54.34	4.46	4.64
20	60	40	3.31	7.84	0.41	0.016	0.064	61.00	26.05	448.39	52.68	4.49	4.61
25	30	40	3.27	7.58	0.42	0.018	0.062	48.50	27.62	296.20	47.13	4.86	4.92
25	45	40	3.34	7.94	0.42	0.017	0.063	54.35	24.78	410.17	53.67	4.50	4.67
25	60	40	3.38	8.29	0.41	0.015	0.064	58.64	23.22	340.12	50.51	4.59	4.65
30	30	40	3.33	7.77	0.40	0.017	0.062	42.17	24.31	260.42	45.43	4.94	5.03
30	45	40	3.39	8.16	0.40	0.015	0.063	49.05	20.79	403.59	50.50	4.82	4.88
30	60	40	3.41	8.41	0.39	0.015	0.065	51.57	19.14	326.27	49.30	4.78	4.84

^1^ SDs—storage days; TT—treatment temperature; TA—titrable acidity; TSSs—total soluble solids; BI—browning index; CI—cloudiness index; AA—ascorbic acid; TVC—total viable count; TPC—total phenolic content; AO—antioxidant activity; TFC—total flavonoid content; YM—yeast and mold count.

**Table 4 foods-13-00497-t004:** The effects of pasteurization and thermosonication frequency on the microbial count of bor-thekera juice ^1^.

Treatment/Condition	TVC (log CFU/mL)	YMC (log CFU/mL)	TVC (log CFU/mL)	YMC (log CFU/mL)
FBTJ	3.95 ± 0.03 ^a^	3.79 ± 0.04 ^a^	3.95 ± 0.03 ^a^	3.79 ± 0.04 ^a^
PBTJ	Not detected	Not detected	Not detected	Not detected
Frequency	33 kHz		44 kHz	
TSBTJ (30 min/30 °C)	3.92 ± 0.03 ^ab^	3.75 ± 0.03 ^ab^	3.88 ± 0.02 ^ab^	3.72 ± 0.04 ^ab^
TSBTJ (45 min/30 °C)	3.90 ± 0.02 ^ab^	3.73 ± 0.03 ^ab^	3.81 ± 0.03 ^b^	3.67 ± 0.03 ^bc^
TSBTJ (60 min/30 °C)	3.88 ± 0.03 ^abc^	3.71 ± 0.02 ^ab^	3.76 ± 0.03 ^bc^	3.59 ± 0.03 ^bc^
TSBTJ (30 min/40 °C)	3.91 ± 0.04 ^abcd^	3.73 ± 0.04 ^ab^	3.86 ± 0.03 ^bcd^	3.67 ± 0.02 ^bc^
TSBTJ (45 min/40 °C)	3.88 ± 0.03 ^abcd^	3.70 ± 0.03 ^ab^	3.73 ± 0.02 ^cde^	3.53 ± 0.03 ^cd^
TSBTJ (60 min/40 °C)	3.86 ± 0.02 ^bcd^	3.69 ± 0.02 ^b^	3.71 ± 0.04 ^de^	3.46 ± 0.04 ^de^
TSBTJ (30 min/50 °C)	3.86 ± 0.01 ^bcd^	3.71 ± 0.01 ^b^	3.84 ± 0.03 ^de^	3.65 ± 0.03 ^de^
TSBTJ (45 min/50 °C)	3.82 ± 0.03 ^cd^	3.69 ± 0.03 ^b^	3.74 ± 0.02 ^e^	3.51 ± 0.02 ^e^
TSBTJ (60 min/50 °C)	3.81 ± 0.03 ^d^	3.67 ± 0.02 ^b^	3.70 ± 0.03 ^e^	3.44 ± 0.04 ^e^

^1^ Values with different superscript small letters in the same column (a–e) are significantly different (*p* < 0.05) from each other. Mean values of three replicates. FBTJ—fresh bor-thekera juice; PBTJ—pasteurized bor-thekera juice; TSBTJ—thermosonicated bor-thekera juice; TVC—total viable count; YMC—yeast and mold count.

**Table 5 foods-13-00497-t005:** Predicted outputs for thermosonicated bor-thekera juice at 44 kHz from ANN modeling ^1^.

SDs	Time	TT	PH	TSSs	TA	CI	BI	AO	AA	TPC	TFC	TVC	YM
0	30	40	3.05	6.83	0.48	0.020	0.062	76.61	43.62	555.86	75.73	3.99	3.89
0	45	40	3.09	6.95	0.47	0.022	0.057	77.45	42.91	564.90	82.61	3.91	3.76
0	60	40	3.14	7.10	0.45	0.023	0.053	78.14	41.98	572.32	87.44	3.85	3.66
1	30	40	3.06	6.86	0.48	0.020	0.062	76.03	43.33	550.44	73.21	4.01	3.93
1	45	40	3.10	6.98	0.47	0.022	0.057	76.95	42.52	560.39	80.63	3.92	3.79
1	60	40	3.15	7.14	0.45	0.023	0.053	77.73	41.48	568.63	86.13	3.86	3.69
3	30	40	3.07	6.91	0.48	0.019	0.062	74.67	42.63	538.00	67.99	4.05	4.04
3	45	40	3.12	7.06	0.46	0.021	0.057	75.79	41.62	549.94	76.08	3.95	3.88
3	60	40	3.17	7.23	0.45	0.023	0.053	76.75	40.34	559.98	82.88	3.88	3.75
5	30	40	3.09	6.98	0.47	0.019	0.062	73.04	41.75	523.36	62.91	4.09	4.15
5	45	40	3.14	7.14	0.46	0.020	0.057	74.39	40.50	537.42	70.98	3.98	3.98
5	60	40	3.20	7.32	0.45	0.022	0.053	75.55	38.98	549.45	78.77	3.90	3.83
7	30	40	3.11	7.05	0.47	0.018	0.062	71.14	40.66	506.51	58.41	4.14	4.27
7	45	40	3.17	7.22	0.46	0.020	0.057	72.71	39.17	522.68	65.77	4.02	4.08
7	60	40	3.22	7.42	0.44	0.021	0.053	74.09	37.41	536.83	73.94	3.93	3.92
10	30	40	3.15	7.17	0.47	0.017	0.061	67.81	38.62	477.47	53.20	4.21	4.44
10	45	40	3.20	7.36	0.45	0.019	0.057	69.69	36.78	496.47	58.70	4.08	4.26
10	60	40	3.25	7.56	0.44	0.020	0.053	71.40	34.72	513.74	66.13	3.98	4.08
15	30	40	3.20	7.41	0.46	0.016	0.061	61.39	34.26	422.96	48.37	4.34	4.68
15	45	40	3.25	7.61	0.44	0.017	0.056	63.54	32.06	444.25	50.91	4.20	4.54
15	60	40	3.29	7.80	0.43	0.019	0.053	65.65	29.89	465.14	55.19	4.07	4.37
20	30	40	3.25	7.65	0.45	0.016	0.061	55.04	29.44	370.05	46.46	4.47	4.84
20	45	40	3.29	7.83	0.43	0.016	0.056	57.01	27.43	389.41	47.43	4.33	4.75
20	60	40	3.33	7.99	0.42	0.018	0.053	59.09	25.66	409.99	49.24	4.19	4.63
25	30	40	3.29	7.87	0.44	0.015	0.061	49.96	25.32	327.86	45.78	4.59	4.94
25	45	40	3.33	8.02	0.42	0.016	0.056	51.43	23.88	342.41	46.13	4.46	4.88
25	60	40	3.35	8.14	0.41	0.017	0.053	53.08	22.72	358.97	46.79	4.32	4.81
30	30	40	3.33	8.05	0.43	0.015	0.060	46.56	22.51	299.19	45.55	4.68	4.98
30	45	40	3.35	8.16	0.42	0.015	0.056	47.49	21.64	308.64	45.67	4.58	4.96
30	60	40	3.37	8.24	0.41	0.016	0.053	48.59	20.98	319.95	45.90	4.45	4.92
R^2^			0.984	0.973	0.970	0.995	0.980	0.992	0.998	0.990	0.970	0.994	0.985
MSE			0.386	0.370	0.118	0.688	0.487	0.188	0.141	0.805	0.591	0.169	0.587
MAE			0.013	0.013	0.004	0.024	0.017	0.006	0.005	0.028	0.020	0.006	0.020

^1^ SDs—storage days; TT—treatment temperature; TA—titrable acidity; TSSs—total soluble solids; BI—browning index; CI—cloudiness index; AO—antioxidant activity; AA—ascorbic acid; TPC—total phenolic content; TVC—total viable count; TFC—total flavonoid content; YM—yeast and mold count; MSE—mean square error; R—coefficient of determination; MAE—mean absolute error.

## Data Availability

Data are contained within the article.

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
