# Peer review of "Optimizing Quality and Shelf-Life Extension of Bor-Thekera (*Garcinia pedunculata*) Juice: A Thermosonication Approach with Artificial Neural Network Modeling"

_foods, 2024, doi:10.3390/foods13030497_

Round 1
Reviewer 1 Report
Comments and Suggestions for Authors
General comments:
- The manuscript reports an interesting study on the potential of thermosonication (TS) to pasteurise bor-thekera juice to preserve the physico-chemical and microbiological quality. The approach is interesting and promising results were obtained, also a different modelling approach was used (artificial neural network modeling). In addition, English writing is good, and the manuscript is easy to follow. However there are some suggestions and modifications to be considered.
- Title: As discussed in one comment, the word “preservation” is probably not the most suitable here, but perhaps shelf-life extension. Some related preservation to studies including inactivation of pathogenic bacteria. Seo maybe consider changing the title as suggested to avoid confusion.
-
- Abstract: One line in the beginning to justify the reason for the study. L18 TS was conducted as pasteurisation treatment? If so, maybe is better to use pasteurised “by” TS and not “and”. L19 Instead of 1 month, indicate the days, depending on the month, different number of days are. L22 Microbial activity is not accurate, how is the microbial activity measured? Microbial growth. Ln27 PBTJ is heat-pasteurised? L28 Microbial safety is wrong here as only total viable counts (TVC), yeast and mould (YM) counts are investigated and they are considered as spoilage groups, nothing related to safety and human health L29 R2 right? L32 Were more juices apart from TSBTJ assessed in the study?
- L55 use passive voice better to avoid using people
- L56 remove etc.
- L57 at room temperature
- L64 Pasteurization is used to inactivate pathogenic microorganisms, being the target of the most heat-resistant. It would be nice to add the term “pathogenic” so as not to confuse with sterilisation treatment
- L69 It seems like novel processing technologies are the only way to tackle the problem of quality improvement, and this is not true. Please rewrite for clarification
- L72 contaminated pathogens, wrong, contaminating
- L72 Italic letters for bacteria
- L73 US has many limitations alone and its use is not considered a “potential substitute” as long treatment time and high energy are required to be seen as an alternative to thermal treatment. It is suggested to change the sense of this paragraph starting by saying that US is not enough in all cases to achieve the target inactivation and being a feasible technology to be upscaled and then link it with TS as a strategy that combines US and heat that can overcome such problems.
- L78, what is a shorter time…? It will depend on the treatment, if you want the make a general statement, maybe you need to include numbers such as time and temperature so as not to confuse.
- L82 cavitation phenomenon needs to be more explained, what is the consequence of the collapse of bubbles and why does it have microbial lethal effects? What happens with the cavitation bubbles when temperature is included in the treatment? They become smaller but the temperature makes more sensible cells and then a higher inactivation is achieved… this needs to be explained
- L86-88 Wordly if not data is shown here
- L101 microbial as well right?
- L111 indicate that is thermal pasteurisation, as TS is also applied as pasteurisation treatment
- L114 Why this combination of treatment and time? Need a justification and better if pathogen bacteria inactivation can justify this, like E. coli O157:H7 in fruit juices. If there is no justification for this combination of treatment and time, the study is not industrially relevant
L119 were treatments performed in dynamic or isothermal conditions? Indicate it, also if the temperature of the product was monitored during the processing
- L128, with which day frequency?
- L130 Better to modify the title or to make another section for the microbial analysis, as it is not a physicochemical analysis
- L142 which procedure, need to be completed this
- L130 more explanation of all the physic-chemical procedures is required, please.
- L160 term safety is incorrect
- L207 The term “non-thermal” may cause a misunderstanding here, as thermosonication is considered a (mild)thermal treatment
- L210 but which can be the reason why TS is found higher pH levels? Further explanation
- L202 In this part it is recommended to discuss the results obtained in this study with works previously done with US or TS with other fruit juices. Also, it is not clear in this part of the paper which is considered “good results”, as there is only a presentation of results, but are they promising with TS? Maybe a further comparison of the TS results with the fresh is needed. It is seen that there are so many results and numbers but nothing relevant to highlight among all numbers, like mentioning significant differences between groups.
- L285 comma before and
- L287 here AA is antioxidant activity (AA) or ascorbic acid?
- L304 in all tables it may be nice to include the significant differences in the results using letters (a,b,c), as in the way that results are presented, they are not very informative
- L352 Further discussion being more specific will be better
- L452 Maybe better illustrate these results in graphs rather than in a table, so much information in Table 3
- L461 change title has also yeast and moulds have been studied and they are not mentioned in the title section
- L506 Table 5 (singular)
- L528 It seems that the observed optimal conditions after ANN were not validated experimentally. It will be important to compare such optimal conditions from the model with the experimental data and results obtained for such conditions. Are the results from the ANN in agreement with the experimental results in terms of all parameters? Mention this in the text.
- L532 If the optimal conclusions are 30 min and 40°C, 30 min is not feasible industrially, as it is a very long processing time… maybe a point to discuss on this can be added
- L535-L627 Are this information of the ANN relevant to the paper? It is shown relevant information? If not, it is suggested to include these graphs and tables in the supplementary data.
- L628 if it is said in line 634 that the microbial inactivation was better in thermal-pasteurised juice, is this in agreement with what is written in line L655?
- L652 Here (in the conclusions section) is the first time that the word “enzymatic extraction” appears, but the work is about thermosonication and physic-chemical and microbial preservation of juices… no extraction with the sonication. It is confusing what you want to conclude here… Please, rewrite and be more clear in what you want to write here.
- L655 Be more specific with functional attributes… functional attributes or better physic-chemical properties. If you want to leave functional attributes, please, mention which.
Comments on the Quality of English Language
General comments:
- The manuscript reports an interesting study on the potential of thermosonication (TS) to pasteurise bor-thekera juice to preserve the physico-chemical and microbiological quality. The approach is interesting and promising results were obtained, also a different modelling approach was used (artificial neural network modeling). In addition, English writing is good, and the manuscript is easy to follow. However there are some suggestions and modifications to be considered before publishing this manuscript.
- Title: As discussed in one comment, the word “preservation” is probably not the most suitable here, but perhaps shelf-life extension. Some related preservation to studies including inactivation of pathogenic bacteria. Seo maybe consider changing the title as suggested to avoid confusion.
-
- Abstract: One line in the beginning to justify the reason for the study. L18 TS was conducted as pasteurisation treatment? If so, maybe is better to use pasteurised “by” TS and not “and”. L19 Instead of 1 month, indicate the days, depending on the month, different number of days are. L22 Microbial activity is not accurate, how is the microbial activity measured? Microbial growth. Ln27 PBTJ is heat-pasteurised? L28 Microbial safety is wrong here as only total viable counts (TVC), yeast and mould (YM) counts are investigated and they are considered as spoilage groups, nothing related to safety and human health L29 R2 right? L32 Were more juices apart from TSBTJ assessed in the study?
- L55 use passive voice better to avoid using people
- L56 remove etc.
- L57 at room temperature
- L64 Pasteurization is used to inactivate pathogenic microorganisms, being the target of the most heat-resistant. It would be nice to add the term “pathogenic” so as not to confuse with sterilisation treatment
- L69 It seems like novel processing technologies are the only way to tackle the problem of quality improvement, and this is not true. Please rewrite for clarification
- L72 contaminated pathogens, wrong, contaminating
- L72 Italic letters for bacteria
- L73 US has many limitations alone and its use is not considered a “potential substitute” as long treatment time and high energy are required to be seen as an alternative to thermal treatment. It is suggested to change the sense of this paragraph starting by saying that US is not enough in all cases to achieve the target inactivation and being a feasible technology to be upscaled and then link it with TS as a strategy that combines US and heat that can overcome such problems.
- L78, what is a shorter time…? It will depend on the treatment, if you want the make a general statement, maybe you need to include numbers such as time and temperature so as not to confuse.
- L82 cavitation phenomenon needs to be more explained, what is the consequence of the collapse of bubbles and why does it have microbial lethal effects? What happens with the cavitation bubbles when temperature is included in the treatment? They become smaller but the temperature makes more sensible cells and then a higher inactivation is achieved… this needs to be explained
- L86-88 Wordly if not data is shown here
- L101 microbial as well right?
- L111 indicate that is thermal pasteurisation, as TS is also applied as pasteurisation treatment
- L114 Why this combination of treatment and time? Need a justification and better if pathogen bacteria inactivation can justify this, like E. coli O157:H7 in fruit juices. If there is no justification for this combination of treatment and time, the study is not industrially relevant
L119 were treatments performed in dynamic or isothermal conditions? Indicate it, also if the temperature of the product was monitored during the processing
- L128, with which day frequency?
- L130 Better to modify the title or to make another section for the microbial analysis, as it is not a physicochemical analysis
- L142 which procedure, need to be completed this
- L130 more explanation of all the physic-chemical procedures is required, please.
- L160 term safety is incorrect
- L207 The term “non-thermal” may cause a misunderstanding here, as thermosonication is considered a (mild)thermal treatment
- L210 but which can be the reason why TS is found higher pH levels? Further explanation
- L202 In this part it is recommended to discuss the results obtained in this study with works previously done with US or TS with other fruit juices. Also, it is not clear in this part of the paper which is considered “good results”, as there is only a presentation of results, but are they promising with TS? Maybe a further comparison of the TS results with the fresh is needed. It is seen that there are so many results and numbers but nothing relevant to highlight among all numbers, like mentioning significant differences between groups.
- L285 comma before and
- L287 here AA is antioxidant activity (AA) or ascorbic acid?
- L304 in all tables it may be nice to include the significant differences in the results using letters (a,b,c), as in the way that results are presented, they are not very informative
- L352 Further discussion being more specific will be better
- L452 Maybe better illustrate these results in graphs rather than in a table, so much information in Table 3
- L461 change title has also yeast and moulds have been studied and they are not mentioned in the title section
- L506 Table 5 (singular)
- L528 It seems that the observed optimal conditions after ANN were not validated experimentally. It will be important to compare such optimal conditions from the model with the experimental data and results obtained for such conditions. Are the results from the ANN in agreement with the experimental results in terms of all parameters? Mention this in the text.
- L532 If the optimal conclusions are 30 min and 40°C, 30 min is not feasible industrially, as it is a very long processing time… maybe a point to discuss on this can be added
- L535-L627 Are this information of the ANN relevant to the paper? It is shown relevant information? If not, it is suggested to include these graphs and tables in the supplementary data.
- L628 if it is said in line 634 that the microbial inactivation was better in thermal-pasteurised juice, is this in agreement with what is written in line L655?
- L652 Here (in the conclusions section) is the first time that the word “enzymatic extraction” appears, but the work is about thermosonication and physic-chemical and microbial preservation of juices… no extraction with the sonication. It is confusing what you want to conclude here… Please, rewrite and be more clear in what you want to write here.
- L655 Be more specific with functional attributes… functional attributes or better physic-chemical properties. If you want to leave functional attributes, please, mention which.
Author Response
we have addressed all the suggestions raised by the reviewer, please see the attached file.

Reviewer 2 Report
Comments and Suggestions for Authors
Comments and Suggestions for Authors are in attach file

Minor corrections are necessary in the form of grammatical errors and misspelled words
Author Response

(The authors gave the same response as above.)

Reviewer 3 Report
Comments and Suggestions for Authors
This study focused on evaluating the quality of pasteurized and thermosonicated Bor-thekera (Garcinia pedunculata) juices during one month of storage at 4°C. Various parameters such as pH, acidity, soluble content, antioxidant activity, phenolic and flavonoid content, ascorbic acid, cloudiness, browning index, and microbial activity were analyzed and compared with fresh Bor-thekera juice. A multi-layer artificial neural network (ANN) was used to model. Thermosonicated juice (TSBTJ) retained more nutritional attributes compared to pasteurized juice (PBTJ) and demonstrated satisfactory microbiological safety. This work is interesting but there are some points that should be considered:
Please don’t use the same words for title keywords.
2.5.2- 2.6.3: Give detail please.
Figs. 5-9: Their quality is not good. Please don’t use the figure that you directly took from the MATLAB software.
When I read title, I expected optimisation problem. But there is no optimization only modelling work.
Comments on the Quality of English LanguageModerate editing of English language required.
Author Response

(The authors gave the same response as above.)

Round 2
Reviewer 1 Report
Comments and Suggestions for Authors
All suggestions and comments were implemented, and the manuscript can be accepted in the current form. Congratulations to the authors.
Comments on the Quality of English Language
Minor improvements could be done, but I leave it to the authors. I as a reviewer am proofing the scientific quality.
Reviewer 2 Report
Comments and Suggestions for Authors
The results are further clarified and any ambiguities are clarified
As such it is acceptable for publication
Reviewer 3 Report
Comments and Suggestions for Authors
Authors have revised the comments I raised.
Comments on the Quality of English LanguageMinor editing of English language required.